# Synthesis and Biological Evaluation of Oleanolic Acid Derivatives as Selective Vascular Endothelial Growth Factor Promoter i-Motif Ligands

**DOI:** 10.3390/ijms22041711

**Published:** 2021-02-08

**Authors:** Huang Zeng, Shuangshuang Kang, Yu Zhang, Ke Liu, Qian Yu, Ding Li, Lin-Kun An

**Affiliations:** 1School of Pharmaceutical Sciences, Sun Yat-sen University, Guangzhou 510006, China; zenghuang8006@163.com (H.Z.); kangshsh@163.com (S.K.); zhangy553@mail2.sysu.edu.cn (Y.Z.); ke.liu.19@ucl.ac.uk (K.L.); yuxi024@163.com (Q.Y.); 2Guangdong Provincial Key Laboratory of New Drug Design and Evaluation, Guangzhou 510006, China

**Keywords:** vascular endothelial growth factor, oleanolic acid, i-motif, gene transcriptional regulation, antiproliferation, apoptosis

## Abstract

Vascular endothelial growth factor (VEGF) is an angiogenic growth factor and plays a key role in tumor progression. The C-rich DNA sequence of *VEGF* promoter can form i-motif structure, which is a potential target for the development of novel anticancer agents. However, there is a limited number of chemotypes as the selective ligands of *VEGF* promoter i-motif, which leaves much room for development. Herein, we report the discovery of the natural oleanolic acid scaffold as a novel chemotype for the development of selective ligands of *VEGF* i-motif. A series of oleanolic acid derivatives as *VEGF* promoter i-motif ligands were synthesized. Subsequent evaluations showed that **3c** could selectively bind to and stabilize *VEGF* promoter i-motif without significant binding to G-quadruplex, duplex DNA, and other oncogene i-motifs. Cell-based assays indicated that **3c** could effectively downregulate *VEGF* gene transcription and expression in MCF-7 cells, inhibit tumor cells proliferation and migration, and induce cancer cells apoptosis. This work provides evidence of *VEGF* promoter i-motif as an anticancer target and will facilitate future efforts for the discovery of oleanolic acid-based selective ligands of *VEGF* promoter i-motif.

## 1. Introduction

Vascular endothelial growth factor (VEGF) is an angiogenic growth factor, which plays a key role in the process of new blood vessel formation through sprouting or splitting from pre-existing vessels, resulting in stimulation of proliferation, migration, survival, and permeability of endothelial cells [1,2,3]. VEGF is commonly overexpressed in a variety of cancer cells and associated with tumor progress and viability rate. VEGF receptor and VEGF proteins have been targeted by small molecule inhibitors and monoclonal antibodies for clinically beneficial effect against a variety of tumors [4,5,6]. However, none of these strategies could affect VEGF overexpression. Therefore, development of strategy for downregulating VEGF overexpression in tumor cells could possibly provide a new approach for anti-cancer therapy.

I-motif structure consists of two parallel duplexes combined in an antiparallel manner via intercalating hemiprotonated cytosine^+^–cytosine base pairs (C^+^–C). It has been previously suggested that i-motif structure can be formed only under acidic condition in vitro. However, recent studies have shown that i-motif can exist under neutral or physiological pH under molecular crowding conditions or in the presence of some cations [7,8]. I-motif structure might play important regulatory roles in biological processes, such as DNA replication, gene transcription, and maintenance of genomic stability [9]. Some i-motif structures in oncogene promoters have been indicated for regulating gene transcriptions, including *PDGFR-β*, *c-myc*, *BCL2*, and *KRAS* [10,11,12,13,14]. It has been shown that small molecules **IMC-48** and **PBP1** (Figure 1) interacted with *BCL2* gene promoter i-motif resulting in up-regulation of gene transcription [12,15], while **B19** and **a9** (Figure 1) interacted with *c-myc* gene promoter i-motif causing downregulation of gene transcription [14,16]. These findings suggested that i-motif structure could be targeted for regulating gene transcription.

The proximal promoter region of *VEGF* gene has guanine-rich and cytosine-rich strands (−85 to −50 bp relative to the initiation of transcription), which contains at least three Sp1 binding sites and controls about 90% VEGF expression. The polyG/polyC strands on this tract are very dynamic and capable of forming non-B-DNA conformations, characterized as G-quadruplexes and i-motifs, respectively [17]. Several small molecule ligands targeting four-stranded G-quadruplex structures have already been evaluated for their therapeutic potential [18,19,20]. However, the selective binder to *VEGF* promoter i-motif has not been developed so far.

It has been shown that steroids **IMC-48** and **IMC-76** could selectively bind to i-motif structure without significant interaction to G-quadruplex [12]. It is possible that the selectivity of these polycyclic structures was due to their lack of polyaromatic planar chromophore leading to decreased π–π stacking interaction to G-quartet [15,21,22]. However, their binding affinity and stability to i-motif structure is not significant, which requires further improvement [21,23,24,25]. Natural products and their derivatives are an important source for the discovery of lead compounds in drug discovery and development [26]. In our initial screening from our compound libraries, we found that oleanolic acid derivatives OA-His [27] (Appendix A) could bind to and stabilize *VEGF* gene promoter i-motif (Appendix A). However, the binding affinity of OA-His to *VEGF* promoter i-motif was still not strong enough (*K*_D_ = 13.3 μM) and the stabilization effect was not significant enough (Δ*T*_m_ = 6.2 ℃), which prompted us to synthesize the triterpene oleanolic acid derivatives.

The basic conformational characteristics of i-motif contain cytosine^+^-cytosine base pairs, loops, grooves and phosphate backbone. The lack of four-stranded planar guanine quartets in i-motif is one of the evidential features distinguished from G-quadruplexes. Based on these characteristics, we hypothesized that the ligand without large planar aromatic scaffold could have selective interaction with i-motif structure versus G-quadruplex structure. Herein, we designed and synthesized three types of oleanolic acid derivatives. To evaluate the effect of the side chains at C-28 on the binding affinity to *VEGF* i-motif structure, Type I compounds were synthesized as shown in Scheme 1 (**3a–3q**) and Scheme 2 (**4a–4o**). To evaluate the effect of A-ring on the binding affinity, the opened A-ring Type II compounds with a side chain were also synthesized as shown in Appendix A (**8a–8h**). To evaluate the effect of the side chain on A-ring on the binding affinity, type III compounds with various amide side chains at C-2 position were synthesized as shown in Appendix A (**10a–10d**). The synthesized compounds were initially screened by using surface plasmon resonance (SPR) assay. The active compounds with the *K*_D_ value less than 50 μM were further determined by thiazole orange (TO) displacement assay and circular dichroism (CD) melting assay. The further experiments in vitro and in cells suggested that compound **3c** could selectively bind to *VEGF* promoter i-motif, resulting in downregulation of *VEGF* gene transcription and proliferation inhibition of tumor cells.

## 2. Results

### 2.1. Chemistry

The type I derivatives were synthesized as shown in Scheme 1 and Scheme 2. Compounds **3a–3q** were synthesized via condensation reaction of oleanolic acid (1) with diverse amine side chains following previously reported procedure (Scheme 1) [28]. The alkyne 3a could further react with various azide materials through copper (I) catalyzed click chemistry to give the target compounds **4a–4p** (Scheme 2) [29].

As shown in Appendix A, oleanolic acid reacted with CH_3_I to give the methyl oleanolate 5, which was oxidized with Jones reagent to give the ketone intermediate 6 [30]. Treatment of compound **6** with m-chloroperoxybenzoic acid produced 3,4-lactone-12,13-epoxy intermediate 7. The stereochemistry of this epoxy group was determined through a NOE correlation between H-12 (*δ* 3.17) and H-18 (*δ* 2.00), suggesting an *α*-configuration. The reaction of intermediate 7 with amines mainly gave the type II compounds **8a–8h**. The NOE spectrum revealed a correlation between the signals of H-13 (*δ* 2.66) and H-26 (*δ* 0.96), suggesting a *β*-configuration for H-13, which have more stable trans junction between rings C and D [31,32].

As shown in Appendix A, intermediate 6 could be reacted with Stiles’ reagent and then treated with DDQ to yield the acid 9 [30]. The type III compounds **10a–10d** were prepared via condensation reaction of 9 with amines.

Finally, 44 oleanolic acid derivatives were synthesized and 40 of them were new compounds (**3c–3q**, **4a–4m**, **8a–8h**, **10a–10d** were newly synthesized; **3a** [29], **3b** [33], **4n** [29], **4o** [29] have been previously reported). The structures of the new derivatives were characterized through NMR and HRMS spectra. The purity of all derivatives for biological evaluation was assessed through evaporative light scattering detector (ELSD)/high performance liquid chromatography (HPLC) method and determined to be more than 95%.

### 2.2. Initial Screening

SPR experiments were carried out to investigate the binding affinities of the synthesized compounds to *VEGF* promoter i-motif, *VEGF* promoter G-quadruplex and the duplex DNA. The oligomer Py24 and Pu24 were used as the *VEGF* i-motif and G-quadruplex forming sequence respectively (Appendix A). As shown in Table 1, thirteen active compounds were found to have binding affinity to *VEGF* i-motif with *K*_D_ values ranging from 1.95 to 31.3 μM, and all tested compounds had no obvious binding affinity to *VEGF* G-quadruplex structure and duplex DNA. The compounds **3c**, **3h**, and **3i** with a long and flexible polyamine chain at C-28 position had good binding affinity to *VEGF* i-motif with the *K*_D_ values of 1.95 (Figure 2A), 3.09 and 5.01 μM, respectively. In comparison, compounds **3j–3q** with relatively rigid side chain did not show binding affinity. In addition, it seems that the hydroxyl terminal functional group of the side chain at C-28 also showed important role on the binding affinity to *VEGF* i-motif structure. For example, compounds **3b** and **4o** had *K*_D_ values of 11.4 and 12.0 μM, respectively. Besides, most type II compounds with A-ring opened and modified derivatives had no obvious binding affinity to *VEGF* i-motif. Only two A-ring opened derivatives, **8b** and **8c**, showed moderate interaction with *K*_D_ values of 31.3 and 21.1 μM, respectively; **10a–10d** did not show obvious interaction to *VEGF* i-motif.

To confirm their binding affinity, the thirteen active compounds **3b–3e**, **3h**, **3i**, **4a**, **4b**, **4k**, **4m**, **4o**, **8b**, and **8c** were selected and further assessed through fluorescent intercalator displacement (FID) assays using thiazole orange. Relative TO displacement ratio (%) was determined at a fixed Py24 concentration (1 μM) with or without the tested compound at a fixed concentration (5 μM). As shown in Table 1, the displacement ratios were basically consistent with the *K*_D_ values found in SPR assays. Among them, **3c** and **3i** have significant displacement ratios of 67% and 51%, respectively.

An ideal i-motif ligand should possess two essential features: excellent i-motif binding affinity and excellent i-motif stabilizing ability. In order to investigate the thermodynamic stabilization effect of these thirteen active derivatives on *VEGF* i-motif, CD melting experiment was also performed. As shown in Figure 2B and Appendix A, with the addition of five equivalents **3c**, the melting temperature (*T*_m_) of Py24 enhanced up to 19.1 ℃, indicating that **3c** had significant stabilizing ability to *VEGF* promoter i-motif. Besides, the compound **3i** had moderate Δ*T*_m_ values (7.0 °C, Table 1). It has been reported that the polyamines could stabilize DNA secondary structures [34,35]. The significantly higher *ΔT*_m_ values of **3c** than the other derivatives promoted us to investigate whether the polyamine side chain (1,5,8,12-tetraazadodecane) of **3c** could stabilize Py24 individually. The results showed that no significant change of *T*_m_ value of Py24 was observed with the addition of 1,5,8,12-tetraazadodecane (Appendix A), which indicated that the thermal stability effect of **3c** on Py24 was dependent on the properties of overall molecule structure. Further CD melting experiments indicated that **3c** had weak effect on *VEGF* G-quadruplex and other promoter i-motif structures (*BCL2*, *KRAS*, *c-kit*, *PDGFR-β*) with *ΔT*_m_ values ranging from 0 to 2.6 ℃ (Appendix A), indicating **3c** could selectively stabilize *VEGF* i-motif structure.

Based on these results, compound **3c** was selected for the subsequent biological experiments.

### 2.3. ***3c*** Could Selectively Bind to VEGF i-Motif Structure

To confirm the interaction of **3c** with *VEGF* i-motif, the microscale thermophoresis (MST) experiment was conducted. After incubation of **3c** with 5′-FAM-labeled *VEGF* promoter C-rich oligomers (FPy24), MST analysis was performed and *K_D_* value was determined to be 1.14 μM (Figure 2C), which was consistent with the *K*_D_ value of 1.93 μM determined by using SPR assay. The interactions of **3c** to *VEGF* G-quadruplex, Duplex DNA and other oncogene promoter i-motif structures, such as *BCL2*, *KRAS*, *c-kit*, *PDGFR-β*, were also determined by using MST. As shown in Figure 2D and Appendix A, the interactions of **3c** to the G-quadruplex, duplex DNA and other oncogene promoter i-motifs were relatively weaker with the *K*_D_ values ranging from 14.4 to 95.1 μM.

The binding affinity and selectivity of **3c** was further investigated by using TO displacement and the concentrations of **3c** required to displace TO by 50% (DC_50_) from DNA secondary structures were determined. As shown in Figure 2E,F, **3c** could effectively displace the bound TO from pre-folded Py24 with DC_50_ of 3.19 μM. In comparison, **3c** exhibited lower affinity for Pu24, *BCL2*, *KRAS*, *c-kit,* and *PDGFR-β* with DC_50_ ranging from 8.01 to 16.1 μM. The results of MST and TO experiments suggested that **3c** had binding selectivity to *VEGF* promoter i-motif.

The interaction mode was investigated by using ESI-MS experiment. The ESI-MS spectra showed that the peak at *m/z* 7794.2 corresponding to the mass weight of Py24-3c adduct was observed after annealing *VEGF* promoter i-motif forming sequence Py24 with **3c** at pH 5.0 (Figure 3A), indicating that **3c** could bind to *VEGF* promoter i-motif with stoichiometry of 1:1. Conversely, the peak for Py24-3c adduct was not observed under neutral annealing condition (Appendix A). The interaction mode of **3c** with *VEGF* promoter i-motif was further studied by using isothermal titration calorimetry (ITC) experiment. The negative value of *ΔH* indicated that **3c** interacted with *VEGF* i-motif possibly via van der Waals interaction and hydrogen bond. The negative value of Gibbs free energy (Δ*G*) indicated that their binding was a spontaneous process. The binding enthalpy was well fitted with a 1:1 binding mode, and the *K*_D_ value was calculated to be 1.13 μM (Figure 3B).

### 2.4. ***3c*** Could Induce the Formation of VEGF i-Motif

To evaluate the ability of **3c** to induce the formation of *VEGF* promoter i-motif, CD titration experiment was conducted. The CD spectra of the oligomer Py24 showed a positive peak at 288 nm and a negative peak at 265 nm under acidic condition (pH ≤ 5.5), indicating the formation of i-motif structure (Appendix A). Under the conditions of pH > 6.0, the positive peak at 288 nm decreased significantly, indicating disruption of i-motif structure. Upon the addition of **3c** at pH 6.0, the CD spectra exhibited a dose-dependent enhancement of the positive peak (~288 nm) along with a red shift from 276 nm to 288 nm, implying that **3c** could induce the formation of *VEGF* promoter i-motif (Figure 4A). Conversely, **3c** could not induce the formations of *VEGF* G-quadruplex and other promoter i-motifs, including *BCL2*, *KRAS*, *c-kit*, *PDGFR-β* (Appendix A).

Fluorescence resonance energy transfer (FRET)–quenching experiment was also conducted to confirm the induction of **3c** to the formation of *VEGF* i-motif. The *VEGF* gene promoter i-motif forming sequence containing a FAM (6-carboxy-fluorescein) label at 5′ and a TAMRA (5-carboxytetramethylrhodamine) label at 3′ (FPy24T, Appendix A) was used. As shown in Figure 4B, the fluorescence quenching of FPy24T solution was obviously found with the titration of **3c**, implying that **3c** could induce the formation of *VEGF* i-motif structure.

^1^H NMR titration experiment was also performed to study the effect of **3c** on the secondary structure transformation of oligomer Py24. As shown in Figure 4C, at pH 6.0 condition, species corresponding to a duplex/hairpin and an i-motif were both observed in ^1^H NMR spectra of Py24. The imino protons at 15–16 ppm are characteristic of hemiprotonated C^+^–C base pairs in an i-motif, while the imino peaks at 13 ppm are characteristic of Watson–Crick base pairs in a duplex or hairpin conformation [12]. In comparison, at pH 5.0 condition, signal peaks for i-motif species increased while signal peaks for hairpin/duplex species decreased, implying a shift of the equilibrium from hairpin/duplex to i-motif structure, which was consistent with the CD assay results (Appendix A). Upon addition of **3c**, the equilibrium shifts from hairpin/duplex to i-motif structure were observed in a dose-dependent manner (Figure 4C). With the addition of 10 equivalents of **3c**, no signal (∼13 ppm) for hairpin/duplex species was observed.

The above results strongly supported that **3c** could significantly induce the formation of *VEGF* promoter i-motif structure.

### 2.5. ***3c*** Could Downregulate VEGF Gene Transcription and Protein Expression

To study the effect of **3c** on the gene transcription in cells, dual-luciferase reporter assay was performed. We cloned the wide-type (Wt-VEGF), devoid (Del-VEGF), and mutant (Mut-VEGF) sequences of *VEGF* corresponding upstream i-motif elements into pGL 4.10 vector respectively (Figure 5A and Appendix A), and transiently transfected them into MCF-7 cells followed by the treatment with **3c** for 24 h. As shown in Figure 5B, the luciferase activity was found decreased in a dose-dependent manner for the plasmid containing wild-type *VEGF* promoter i-motif forming sequence. In contrast, no significant decrease of luciferase activity was observed for the plasmid containing devoid and mutated *VEGF* promoter i-motif forming sequence.

Quantitative real time PCR assay was also conducted to analyze the transcription levels. As shown in Figure 5C, after treatment of MCF-7 cells with **3c** for 48 h, *VEGF* mRNA level was found to be reduced significantly in a dose-dependent manner. In contrast, **3c** had no significant effect on the transcription of other oncogenes, including *BCL2*, *KRAS*, *c-kit,* and *PDGFR-β*.

The effect of **3c** on VEGF protein expression was determined by using Western blot experiment. As shown in Figure 5D, **3c** could reduce VEGF protein expression in a dose-dependent manner in MCF-7 cells. In contrast, **3c** had no significant effect on expression of other oncogene proteins, including *PDGFR-β*, *c-kit,* and *BCL-2* (Appendix A).

These results indicated that **3c** could selectively interact with *VEGF* promoter i-motif, resulting in downregulation of *VEGF* gene transcription and protein expression in cancer cells.

### 2.6. ***3c*** Could Induce Apoptosis and Inhibit Proliferation and Metastasis of Cancer Cells

Annexin–fluorescein isothiocyante (FITC)/propidine iodide (PI) double-staining flow cytometry was carried out to analyze cell apoptosis induced by **3c** (Figure 6A). After being treated with **3c**, the percentages of both early and late apoptotic MCF-7 cells significantly increased. Total 39.6% of apoptotic cells were found after being treated with 6 μM of **3c**.

Since **3c** could downregulate *VEGF* gene transcription and expression in cancer cells, the effects of **3c** on the proliferation and migration of cancer cells were determined. The methyl thiazolyl tetrazolium (MTT) colorimetric assay indicated that **3c** exhibited dose-dependent anti-proliferation activities in MCF-7 cells with its IC_50_ values (the concentration for 50% inhibition) determined to be 6.5 μM, as shown in Appendix A. The colony formation assay further demonstrated that the ability of MCF-7 cells to form colonies was significantly decreased in a dose-dependent manner upon treatment with increasing concentration of **3c** (Figure 6B). In addition, the scrape assay indicated that MCF-7 cell metastasis was slowed down in a dose-dependent manner after being treated with **3c** for 48 h and 96 h period (Figure 6C).

These results indicated that **3c** could inhibit the proliferation and migration of cancer cells and induce cancer cells apoptosis possibly through downregulating *VEGF* gene transcription and translation.

### 2.7. Molecular Modeling

Molecular docking studies were performed to understand possible interactions of **3c** with *VEGF* promoter i-motif structure using the MOE program. Because neither the NMR nor the X-ray crystallographic structure of *VEGF* promoter i-motif has been reported yet, the telomeric i-motif structure (PDB ID:1ELN) [36] was used as a model structure for analyzing possible binding interaction of **3c** with i-motif. As shown in Figure 7, the docking results revealed that the pentacyclic triterpenoid backbone of **3c** possibly lay on the i-motif groove space. The polyamine side chain of **3c** might be positively charged in physiological conditions, and form electrostatic interactions toward the negatively charged phosphate diester backbone of DNA. The terminal amino side chain and the hydroxyl group of **3c** could form two important H-bonds with phosphate backbone with the distances of 2.3 and 3.1 Å, which facilitated **3c** twining around the i-motif structure. This could help to rationalize the significant thermal stability effect of **3c** on Py24. Furthermore, in order to compare the binding mode of **3c** with i-motif and G-quadruplex, the molecular docking of **3c** with *VEGF* promoter G-quadruplex (PDB ID: 2M27) [37] was also performed. The results showed that pentacyclic triterpenoid backbone of **3c** was unable to form the π–π stacking interaction with the planar of G-quadruplex (Appendix A). With the exception of an H-bond forming between the polyamine side chain and dT22 site of phosphate diester backbone, **3c** couldn’t form effective interaction with G-quadruplex. These results explained the reason of the selective interaction of **3c** with i-motif versus G-quadruplex.

## 3. Conclusions

In summary, a series of oleanolic acid derivatives as *VEGF* promoter i-motif ligands were synthesized. The SPR experiment, TO displacement experiment and CD melting experiment were carried out to study their binding affinity and stabilizing ability to *VEGF* i-motif promoter. Among them, **3c** was identified as a promising ligand, which could selectively bind to and stabilize *VEGF* promoter i-motif structure, without significant interaction to *VEGF* G-quadruplex, duplex DNA and other oncogene promoter i-motifs, including *BCL2*, *c-kit*, *KRAS,* and *PDGFR-β* i-motifs. Cell-based assays indicated that **3c** could effectively downregulate *VEGF* gene transcription and translation, and inhibit tumor cells proliferation and migration, and promote cancer cells apoptosis. Therefore, **3c** could be used as a tool compound for the investigation on the role of *VEGF* promoter i-motif in regulating oncogene transcription, and as a promising lead compound for further development for anticancer therapy. This work proved that the natural oleanolic acid scaffold represents a chemotype for selective ligands of *VEGF* i-motif and offered a new anticancer strategy with *VEGF* promoter i-motif as a new target.

## 4. Materials and Methods

### 4.1. General Materials

All chemicals and starting materials were purchased from commercial sources, which were analytical grade without further purification unless otherwise specified. NMR spectra were recorded on a Bruker BioSpin GmbH 400 MHz or 500 MHz spectrometer using TMS (tetramethylsilane) as the internal standard. High-resolution mass spectra (HRMS) were recorded on Shimadzu LCMS-IT-TOF of MAT95XP mass spectrometer (Thermo Fisher Scientific, Waltham, MA, USA). All oligomers/primers used in this study were purchased from Invitrogen or Sangon, as shown in Appendix A. CD experiments were performed on a Chirascan circular dichroism spectrophotometer (Applied Photophysics, Leatherhead, UK). Fluorescence experiments were performed on Fluoromax-4 luminescence spectrophotometer (Horiba Jobin-Yvon, Paris, France). ITC experiments were performed using a MicroCal VP-ITC instrument (GE Healthcare, Northampton, MA, USA). For obtaining i-motif structures, C-rich oligonucleotides were annealed in 1× BPES buffer (30 mM KH_2_PO_4_, 30 mM K_2_HPO_4_, 1 mM EDTA, 100 mM KCl) with different pH at 95 °C for 5 min, and then cooled to room temperature. For obtaining G-quadruplex, oligonucleotides were annealed in 20 mM Tris–HCl buffer containing 100 mM KCl (pH 7.4) by heating at 95 °C for 5 min followed with gradual cooling to room temperature.

### 4.2. Synthesis

All synthetic procedures, yields, and physical and spectroscopic data for the compounds are included in the Appendix A.

### 4.3. Surface Plasmon Resonance (SPR) Assay

SPR assay was performed on a ProteOn XPR36 Protein Interaction Array system (Bio-Rad Laboratories, Hercules, CA, USA) using a NeutrAvidin-coated GLH sensor chip. For immobilization, all DNA samples were biotinylated and attached to a reptavidin-coated sensor chip. 5′-Biotin labeled Py24 were diluted to 1 μM in running buffer (20 mM 2-(4-morpholino)ethanesulfonic acid, pH 5.5, 100 mM KCl and 0.05% Tween-20), and 5′-biotin Pu24 and 5′-biotin duplex DNA were diluted to 1 μM in running buffer (Tris-HCl 20 mM, pH 7.4, 100 mM KCl). The DNA samples were then captured (1000 RU) in flow cells, and a blank cell was set as a control. All ligands were prepared with the running buffer through serial dilutions from stock solution (10 mM in DMSO). Ligands of different concentrations were injected at a flow rate of 30 μL/min for 120 s of association phase, followed with 120 s of dissociation phase at 25 °C. The GLH sensor chip was regenerated with short injection of 50 mM NaOH between consecutive measurements. The final graphs were obtained by subtracting blank sensorgrams from the i-motif, G-quadruplex, or duplex sensorgrams. Data was analyzed by using ProteOn manager software.

### 4.4. Microscale Thermophoresis (MST) Assay

The thermophoresis movements of the fluorescently labeled nucleic acids and **3c** complexes were analyzed by monitoring the fluorescence distributions inside the capillary by using the NT.115 MST instrument (NanoTemper, München, Germany). The concentration of DNA was held constant at 1 μM. The compound was diluted at 1:1 from 100 μM for 16 times. The samples were loaded into standard-treated MST-grade glass capillaries. The intensities of the light-emitting diode (LED) and laser were set as 40% and 40%, respectively. Data was analyzed by using NT Analysis 1.4.23 software (NanoTemper, München, Germany).

### 4.5. TO Displacement Assay

I-motif or G-quadruplex oligonucleotides solution (1 μM, 100 μL) was added to the cuvette, followed with addition of thiazole orange (2 μM). The spectrum of the oligonucleotide with titration of **3c** was recorded based on fluorescence emission (*λ*_ex_ = 480 nm).

### 4.6. ESI-MS Assay

The oligomer Py24 (5 µM) was annealed in the presence or absence of **3c** in 1 × BPES buffer (10 mM KH_2_PO_4_, 10 mM K_2_HPO_4_, 1 mM EDTA, 100 mM KCl, pH 5.0 or pH 7.5) at 95 °C for 5 min. The solution was gradually cooled to room temperature, and detected by using an electrospray ionization (ESI) mass spectrometer.

### 4.7. Isothermal Titration Calorimetry Assay

The oligomer Py24 was diluted from stock to the indicated concentration (0.80 mM) in 1 × BPES buffer at pH 5.0 and then annealed by heating at 95 °C for 5 min. The solution was gradually cooled to room temperature to generate i-motif structure. The oligomer Py24 was titrated into **3c** (0.08 mM) at 30 °C in 10 μL injections with a spacing of 300 s between injections. The integrated heat data were corrected for the heat of dilution and blank effects. The binding isotherms were fitted with the one-site binding model incorporated into the MicroCal Origin VPITC software.

### 4.8. Circular Dichroism Spectroscopy

A quartz cuvette with 4 mm path length was used for the spectra recorded over a wavelength range of 230–350 nm at 1 nm bandwidth. The oligomer Py24 was diluted from stock to the required concentration (1 μM) in 1 × BPES buffer at pH 5.0 or 6.0 and then annealed by heating at 95 °C for 5 min. The solution was gradually cooled to room temperature, and incubated with **3c** at 37 ℃ for 1 h. Spectra were recorded three times, averaged, smoothed, and baseline corrected to remove signal contribution from buffer. The data was analyzed by using Origin 9.0 (OriginLab, Northampton, MA, USA).

For CD melting assays, thermal melting was monitored at 230–350 nm with a 1 nm bandwidth. CD melting assays were performed at a fixed DNA concentration (2 μM) with or without **3c** in relevant buffer. The data were recorded at intervals of 5 °C over a range of 20–95 °C with a heating rate of 1 °C/min.

### 4.9. Fluorescence Resonance Energy Transfer (FRET) Assay

Oligomer Py24 with a 5′-end FAM-fluorophore and a 3′-end TAMRA are provided in Appendix A. The fluorescence of a fixed concentration (0.1 μM) of oligonucleotide was tested (*λ*_ex_ = 480 nm) with the titration of **3c**.

### 4.10. NMR Titration Experiment

The DNA oligonucleotide was purchased from Sangon (China). The final NMR samples were prepared in 1×BPES buffer containing 10% D_2_O at pH 5.0 or 6.0. The concentration of DNA sample was set at 1.0 mM. The stock solution of **3c** was dissolved in DMSO-*d*_6_. ^1^H NMR titration experiments were performed on a Bruker DRX-600 MHz spectrometer at 25 °C, and the water signal was suppressed.

### 4.11. Cell Culture

Human breast adenocarcinoma cell line MCF-7 was purchased from China Center for Type Culture Collection in Wuhan. The cell line was maintained in DMEM (Dulbecco’s Modified Eagle’s medium), supplemented with 10% fetal calf serum, 100 U/mL Penicillium, and 100 mg/mL streptomycin at 37 °C in a humidified atmosphere with 5% CO_2_.

### 4.12. MTT Assay

Human breast adenocarcinoma cells MCF-7 were seeded on 96-well plates (5000 cells/well) and treated with **3c** at various concentrations. After 48 h treatment at 37 °C in a humidified atmosphere of 5% CO_2_, 20 μL of 2.5 mg/mL 3-(4,5-dimethyl-2-thiazolyl)-2,5-diphenyl-2H-tetrazolium bromide (MTT) solution was added to each well and further incubated for 4h. The cells in each well were then treated with DMSO (200 μL for each well), and the optical density (OD) was recorded at 570 nm. The cytotoxicity was evaluated based on the percentage of cell survival in a dose-dependent manner regard to the blank. The final IC_50_ values (the concentration for 50% inhibition) were calculated by using the Prism 8 Computer Software (GraphPad, San Diego, CA, USA). All of the experiments were repeated three times.

### 4.13. Colony Formation Assay

MCF-7 cells were subsequently seeded in 6-well culture plates (1000/well) for 24 h pre-culture at 37 °C in a humidified atmosphere with 5% CO_2_, and then treated with **3c** at different concentrations for 9 days. The cells were washed with 1 × PBS and fixed with ice cold methanol for 10 min, followed by the addition of 0.5% crystal violet solution for 30 min to observe the colony formation.

### 4.14. Cell Scrape Assay

MCF-7 cells were subsequently seeded in 6-well culture plates (300,000/well) at 37 °C in a humidified atmosphere with 5% CO_2_. After being pre-cultured for 48 h, a cross-shaped scrape was made through the monolayer Siha cells using a plastic pipet tip, and then the cells were treated with **3c** at different concentrations. The wounded areas were observed and photographed using microscopy after being incubated for the indicated times.

### 4.15. Flow Cytometric Analysis

MCF-7 cells were seeded in 6-well plate (2 × 10^5^/well) and incubated for 24 h at 37 °C in a humidified atmosphere with 5% CO_2_. After being incubated in the presence and absence of **3c** for 48 h, the Siha cells were then washed in PBS, and centrifuged and re-suspended in Annexin V–FITC solution for 15 min at room temperature in dark. After centrifuged for 5 min, the cells were then re-suspended in Annexin V–FITC solution and mixed with PI staining solution for 10 min at 2–5 °C in dark. Then, the cells were analyzed by using flow cytometry with an Epics Elite flow cytometer (Beckman Coulter, Brea, CA, USA).

### 4.16. Dual-Luciferase Reporter Assay

A total of 100 ng of constructed pGL4.10 luciferase plasmid (Promega, Madison, WI, USA) containing *VEGF* corresponding upstream i-motif elements and 100 ng plt-tk Renilla plasmid were transfected into MCF-7 cells by using Lipofectamine 3000 (Invitrogen, Carlsbad, CA, USA). After 6 h, the solution of **3c** was added to the cells. The cells were incubated at 37 °C with CO_2_ for 24 h, and the transfected cells were first washed with ice-cold PBS to reduce the background signals from the medium. Luciferase assays were subsequently performed according to the manufacturer’s instructions using the dual-luciferase assay system (Promega, Madison, WI, USA). After a 3 s delay, secreted luciferase signals were collected for 10 s using a microplate reader (Molecular Devices, Sunnyvale, CA, USA). The quantification was performed using a multimode reader (Molecular Devices, Sunnyvale, CA, USA). The secreted Renilla luciferase activity was normalized to the firefly luciferase activity.

### 4.17. RNA Extraction and Real Time Polymerase Chain Reaction (RT-PCR)

MCF-7 cells were seeded in 6-well plate (2 × 10^5^ cells/well), and incubated for 24 h at 37 °C in a humidified atmosphere with 5% CO_2_. After the cells were incubated in the presence or absence of **3c** for 48 h, cells were harvested, and the RNA was extracted by using HiPure Total RNA Mini Kit (Magen, Guangzhou, China). Quantitative real-time polymerase chain reaction (PCR) was carried out using 2×RealStar SYBR Mixture (Genstar, Guangzhou, China). The results were analyzed on a LightCycler480 II real-time PCR system (Roche, Basel, Switzerland). The oncogene mRNA levels were normalized to β-actin mRNA level of each sample. Results of real-time PCR were analyzed by using the 2^−Δ*C*t^ method.

### 4.18. Western Blot Assay

MCF-7 cells were seeded in 6-well plate (2 × 10^5^ cells/well) and incubated for 24 h at 37 °C in a humidified atmosphere with 5% CO_2_. After being incubated in the presence and absence of **3c** for 48 h, cells were harvested from each well of culture plates and lysed in 200 μL of protein extraction buffer consisting of 1 mM PMSF for 30 min. The suspension was centrifuged at 10,000 rpm at 4 °C for 15 min, and the protein content of supernatant was measured by using bicinchoninic acid (BCA) assay. The same amount of protein for each sample was loaded onto 8% polyacrylamide gel, and then transferred to a microporous polyvinylidene difluoride (PVDF) membrane. Western blotting was performed by using anti-VEGF and anti-β-GAPDH (cell signaling technology) antibodies, as well as horseradish peroxidase-conjugated anti-rabbit secondary antibody. Protein bands were visualized by using chemiluminescence substrate.

### 4.19. Molecular Docking

The telomeric i-motif DNA and *VEGF* G-quadruplex structures were downloaded from Protein Data Bank (1ELN, 2M27). After optimization of the ligand and assigning partial atomic charges, docking calculations were performed with the MOE program. The ligand to be modeled was constructed and optimized by using ChemDraw and saved in SDF file format, and was minimized with the conjugate gradient method using the MMFF94x force field with MMFF94 charges. The process of site finder was performed using MOE software (Chemical Computing Group, Montreal, Canada) to determine the binding sites. Docking runs were carried out and the ligand conformation was chosen based on the docked energy. The final graphs were created by using PyMOL software (Schrödinger, New York, NY, USA). 

## Data Availability

The data presented in this study are available in both article and Appendix A.

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
