# Peer review of "Synthesis and Biological Evaluation of Oleanolic Acid Derivatives as Selective Vascular Endothelial Growth Factor Promoter i-Motif Ligands"

_ijms, 2021, doi:10.3390/ijms22041711_

Round 1
Reviewer 1 Report
This is a nicely conducted study presenting a series of oleanolic acid derivatives with VEGF i-motif selective ligands. As a continuation of the previous work, the authors attempted to test oleanolic acid derivatives (albeit previously oleanolic acid derivatives have been studied as influenza virus inhibitors) to identify potent, selective VEGF i motif ligand to down-regulate VEGF gene transcription to inhibit tumor cells proliferation and migration, and induce cancer cells apoptosis. A series of intriguing, well-grounded hypotheses, based on medchem knowledge, intuition as well as the author's previous experience are clearly presented and experimentally confirmed. Authors were partially successful to discover the moderately potent and selective ligand 3c. The presented oleanolic acid analogs are undoubtedly still a good start of a medchem campaign. The article is well structured, clearly written, and the presented experimental confirmation of the synthesized ligands warrants and delivers valid results. In my opinion, it is suitable for publication in the International Journal of Molecular Sciences once the minor issues listed below are covered appropriately.
- Despite obtaining the moderate and selective ligand 3c, authors have failed to build up SAR study around that analog to establish good SAR guidelines. Would the author comment on this short come?
- The motivation to test oleanolic acid analogs as VEGF i-motif ligands are the previously published very similar steroids IMS-48 and IMS-76 (JACS, 2013). Please add a figure in the introduction section with structures of IMS-48, IMS-76 as well as structures of previously published compounds. (See J. Med. Chem.2020, 63, 17, 9136–9153)
- The text under the discussion topic looks like the conclusion, please provide an appropriate heading.
- Scheme 1, 3a NHCH3CCH should be NHCH2CCH
- Move Schemes 3 and 4 to the supporting material to avoid too much deviation from the synthesis of 3c and its related analogs.
Reviewer 2 Report
The MS describe the synthesis and biological evaluation of oleanolic acid derivatives as selective vascular endothelial growth factor promoter i-motif ligands.
The conclusions are well justified by the experimental data and results presented by the authors.
All experiments were described in detail.
I recommend the acceptance of the MS as proposed by the authors.
Author Response
Thanks for the reviewer's kind and positive comments.

Reviewer 3 Report
In this manuscript, the authors synthesized and evaluated the oleanolic acid derivatives. SPR-based selection and further biological evaluation revealed that derivative 3c selectively bind to and stabilize VEGF promoter i-motif. The cell-based assays are also reported to evaluate the cancer cell proliferation and migration.
Following remarks must be addressed for considering publication.
1. [Schemes 1-4] Definitions of types I, II and III are only described in the main text, not in the Schemes 1-4. For readability, the definitions should be displayed in the Schemes or written in the captions. In addition, synthetic yields should be displayed in the Schemes 1-4.
2. [Page 2, line 91] "TO" should be spell out as thiazole orange.
3. [Page 5, line 129, "44 oleanolic acid derivatives were synthesized and 40 of them were new compounds."] Compound numbers that are newly synthesized should be clarified. References should be added to the previously reported compounds.
4. [Figure 5] The word "histogram" should be changed to "dot plot". Dose dependency of the growth inhibition of MCF-7 cells should be quantitatively evaluated. This reviewer suggests assessing the 50% growth inhibitory concentration by SRB or MTT assay instead of the colony formation assay.
5. [Supporting Information] In the 1H NMR data, the chemical shifts of several protons (1.2-2.4 ppm) are not listed. All the protons must be listed as characterization data.
Minor points
- All the compound numbers should be written as bold fonts.
- Addition of data of OA-His to Table 1 might be effective for improving readability.
- If possible, quality of 1H NMR of 3b should be improved.
